# Social Connectedness, Physical Distancing, and Anxiety in Complying with Shelter-In-Place Orders and Advisories during the Once-In-A-Century COVID-19 Pandemic in the US: A Study of Social Media and Internet Users

**DOI:** 10.3390/ijerph192215282

**Published:** 2022-11-18

**Authors:** Dean Kyne, Candace Robledo, Loren Cliff Clark, Ruby Charak, Meliha Salahuddin, Jay Morrow

**Affiliations:** 1Department of Sociology, College of Liberal Arts, University of Texas Rio Grande Valley (UTRGV), Edinburg, TX 78539, USA; 2Department of Population Health & Biostatistics, School of Medicine, University of Texas Rio Grande Valley (UTRGV), Edinburg, TX 78539, USA; 3Department of Psychology, College of Liberal Arts, University of Texas Rio Grande Valley (UTRGV), Edinburg, TX 78539, USA; 4School of Public Health Austin Campus, University of Texas Health Science Center at Houston (UTHealth), Austin, TX 78701, USA; 5Peter O’Donnell Jr. School of Public Health, University of Texas Southwestern Medical Center, Dallas, TX 75390, USA

**Keywords:** social connectedness, anxiety, shelter-in-place, COVID-19

## Abstract

The severe acute respiratory syndrome coronavirus 2 (SARS-CoV-2), which causes coronavirus disease (COVID-19), was first identified in Wuhan, China, in December 2019. As of 20 October 2020, the virus had infected 8,202,552 people, with 220,061 deaths in US, and in countries around the world, over 38 million people have become infected and over one million have died. The virus usually spreads via respiratory droplets from an infected person. At the time of compiling this paper, while countries around the world are still striving to find a “pharmaceutical intervention (PI)”, including treatments and vaccines, they are left with only “non-pharmaceutical interventions (NPIs)”, such as physical distancing, wearing masks, and maintaining personal hygiene. In the US, all 50 states, the District of Columbia, and five US territories issued mandatory stay-at-home orders between March 1 and 31 May 2020 to lower the risk of virus transmission. This study empirically examined how social connectedness and anxiety interact with shelter-in-place compliance and advisories during the pandemic. The study collected information from 494 adults using an online survey during April and July 2020.

## 1. Introduction

The current and unprecedented once-in-a-century COVID-19 pandemic is drastically changing the world. Nobody suspected that a novel virus of about 100 nanometers in diameter [1], which was first spotted in mid-December 2019 in China, would have such an unprecedented and vast impact on all aspects of life. After the novel virus was named Severe Acute Respiratory Syndrome Coronavirus 2 (SARS-CoV-2) on 7 January 2020 [2], four days later, the virus took the life of its first victim, a 61-year-old man in China. To begin with, the virus showed explosive levels of transmission and lethality. Globalization added fuel to the fire of unprecedented virus transmission as the virus spread from its origin to other countries around the world. Two weeks later, the authorities in China reported a total of 139 known cases and three deaths, and there were confirmed cases in other counties including Japan, South Korea, and Thailand. On 21 January 2020, the novel virus reached US soil as the US Centers for Disease Prevention and Control (US CDC) first recorded a confirmed case in Washington State.

A month later, there were 14 known cases and two deaths in US. Soon after the World Health Organization (WHO) announced the coronavirus outbreak as a pandemic on 11 March, the Trump administration declared a state of national emergency. On 22 March 2020, nine days after a national emergency was declared, the US recorded a total of 19,152 known cases with an alarming record of 445 deaths. In the following consecutive months, coronavirus cases continued to spike to unprecedented levels as the nation’s CDC reported 776,787 known cases and 46,194 deaths on 20 April; 1,540,631 cases and 95,203 deaths on 20 May; and 2,240,215 cases and 119,445 deaths on 20 June. As of 20 October 2020 when this report was under preparation, the US was ranked first in the world with an unprecedented record of 8,202,552 cases and 220,061 deaths among 189 countries/regions around the world [3,4].

The novel virus is transmitted via respiratory droplets from an infected person to those who are in close contact with each other (within about 6 feet) [5]. To combat a pandemic, there are two recommended strategies: pharmaceutical intervention (PI) and non-pharmaceutical intervention (NPI). The first strategy was not immediately available in the early stage of pandemic because it takes time to develop a well-matched pandemic strain vaccine and effective therapeutic drugs [6]. The latter strategy calls for collective actions from individuals and communities to slow the spread of coronavirus transmission [6,7]. They include isolation and treatment, voluntary home quarantine of members of households with confirmed infection, the dismissal of students from school, and closure of childcare programs [6]. While many vaccine developers and researchers are carrying out clinical trials to evaluate the results, [8] the implementation of the NPIs is expected to last until the winter of 2020–2021 [9].

Studies, e.g., [9] found that working from home and sheltering in place under the social distancing orders in fact helps slow the spread of the virus and flatten the curve. Childs et al. shared their findings that long-term moderate social distancing coupled with testing and the isolation of symptomatic individuals could contain the sizable spread of the COVID-19 pandemic in Santa Clara County, California, and it is expected to be even more effective in other locations. However, social distancing is not a simple task for everyone since it contradicts the ultra-social nature of humans [10]. While social distancing is necessary to slow the transmission of the virus, isolation could potentially increase loneliness, which could lead to anxiety, stress, and undesired illnesses [10,11]. However, there are limited studies that focus on the effects of long-term social distancing on mental health [12]. In this study, as a team of interdisciplinary researchers, we empirically investigate how physical distancing during this pandemic plays out with the concepts of social connectedness and anxiety, utilizing sociology, medicine, psychology and environmental social science lenses.

This study is organized as follows. First, we discuss three concepts—social connectedness, family support, and social provisions—and the ways that these correlate with physical distancing during the pandemic. Then, we discuss our hypothesized structural equation (SEM) model of social connectedness, physical distancing, and anxiety (SCPDA) as well as our hypothesis statements. The next section on data and methods describes the study area, survey instruments, data collection procedures, and structural equation modeling. In the results section, we discuss the statistical significance of testing hypotheses, goodness of model fit, residuals of observed variables, and total effects of the study variables. Based on the study’s findings, we conclude that physical distancing interacts with social connectedness and anxiety. We also provide recommendations on possible policies to promote physical distancing in compliance with pandemic stay-at-home or shelter-in-place orders or advisories to minimize potential anxiety.

### 1.1. Physical Distancing

Social distancing refers to “physical distancing of people where in circumstances, where there are crowds, you remove yourself from very close contact [13]”. Social distancing includes teleworking, avoiding crowded bars, restaurants, and theaters, staying at least six feet apart, and avoiding unnecessary trips [13]. Following the guidelines provided by the experts from the White House Coronavirus Task Force (WHCTF) team, all 50 states and the District of Columbia and five US territories issued stay-at-home orders between 1 March and 31 May 2020 as a community mitigation strategy with the primary goal of reducing the spread of coronavirus in the United States [14]. Two underlying reasons for adopting stay-at-home orders as a community mitigation strategy are that they can (1) limit the transmission of the virus through respiratory droplets from an infected person to others who are in close contact and (2) limit person-to-person interactions, which could result in reducing risks of viral transmission [14]. Under the statewide stay-at-home orders, jurisdictions are required to (1) close nonessential businesses, (2) furlough most government and commercial employees, (3) prohibit public events or gatherings, (4) restrict travel orders to leave home except for necessities such as groceries and medical care [15].

While the scientific community was frustrated over the delayed actions of governments in adopting mandatory stay-at-home orders [16], states adopted five types of stay-at-home orders: (1) mandatory for all persons; (2) mandatory only for persons in certain areas of the jurisdiction; (3) mandatory only for persons at increased risk in the jurisdiction; (4) mandatory only for persons at increased risk in certain areas of the jurisdiction; or (5) advisory or recommendation (i.e., nonmandatory) [14]. There were eight jurisdictions that issued only an advisory order or recommendation to stay home, while six jurisdictions did not issue any stay-at-home orders. During the period between 1 March and 31 May 2020, there were 42 states and territories (73% of total counties) that issued mandatory stay-at-home orders. However, studies found Americans stayed at home in every state even before stay-at-home orders were issued from 15 March [17]. Our study focuses on all types of physical distancing, which include observing stay-at-home orders, advisories, or self-imposed sheltering in place.

The stay-at-home order was viewed as a stricter form of social distancing, which includes staying at home, only going out for essential businesses, watching distance of six feet apart, and no gathering in groups [18]. Under the stay-at-home order, while wearing face masks, people could still carry out many of the activities that they did before the pandemic, which include; (1) shopping at grocery, convenience, and warehouse stores, (2) picking up medications at a pharmacy, (3) visiting doctor appointments, (4) going to a restaurant for takeout, delivery, or drive-thru, (5) supporting a friend or family member, (6) taking a walk, riding a bike, hiking, or jogging while maintaining a distance of six feet from others, (7) walking your pets and taking them to a veterinarian, (8) helping others to obtain necessary supplies, and (9) receiving deliveries from businesses [18]. However, people were advised (1) not go to work unless they are essential workers, (2) limit unnecessary visits to friends and family, (3) maintain at least a six feet distance when you need to go out, and (4) not to visit loved ones in hospitals and nursing homes.

The primary purpose of staying at home was to slow down the spread of coronavirus transmission. At the time of completing this manuscript in October 2020, studies, e.g., [19] documented the reduced COVID-19 infection rate in the states where stay-at-home orders were issued and in place. With such encouraging evidence, states that reopened still required wearing masks and observing physical distancing [20]. Our study focuses on physical distancing in relation to social connectedness and anxiety, excluding measuring the effectiveness of reducing coronavirus infection rates.

### 1.2. Anxiety

In the midst of the pandemic, individuals have to cope with anxiety stemming from many unknowns associated with the potential infection from deadly coronavirus [21]. This anxiety could be amplified by the requirements of physical distancing in the stricter form of staying at home and self-imposed isolation, which is against the fundamental ultra-social nature of humans [10]. In addition, fear of losing employment and income adds another layer to the anxiety of individuals during this pandemic [22]. In fact, individuals who were under a stay-at-home order experienced anxiety, financial worry, and loneliness [23]. In one study, self-isolation coupled with economic loss and changes in daily routine was found to have psychological and emotional impacts under stay-at-home orders [24]. Imposing stay-at-home orders and physical distancing can negatively impact mental and physical health in the long term. Several studies documented associations between physical distancing and its association with both physical and mental health before the pandemic [25,26].

During the pandemic, more people felt lonely than ever before. It was found that 61 percent of Americans over the age of 18 years felt lonely, comprising 79 percent of Gen Z-ers, 71 percent of millennials, and 50 percent of baby boomers [27]. In one study, about 70 percent of participants who were older than 70 claimed to never feel lonely, but during the COVID-19 pandemic, feelings of loneliness for the participants increased because of physical distancing and social isolation measures, which interrupted their routine community or church-based social participation and engagement [28,29]. Similarly, some children also felt loneliness and anxiety because of school closures and family trauma, which led to social isolation, stress and mental-health issues [30]. According to a study by Harvard University researchers involving 224 children between 7 and 15 years of age, two-thirds of them had symptoms of anxiety and depression with hyperactivity and inattention from November 2020 to January 2021, a dramatic increase of 30% compared to the situation before the pandemic [30].

### 1.3. Social Connectedness

Social connectedness refers to “feelings of belonging to a group of people and of being close to them” [31]. Before the start of pandemic, the nation faced a high level of loneliness, which is a continuing public health issue [10]. According to a meta-analytic review by Julianne Holt-Lunstad, professor of psychology and neuroscience, social relationships could increase an individual’s likelihood of survival by 50%, and a lack of human connection could have a negative impact on life expectancy that is similar to the effects of smoking 15 cigarettes a day [32]. A strong association between social disconnectedness and physical and mental health outcome was observed [25]. According to a study in US that analyzed over 10 million Google searches, anxiety, which included negative thoughts; sleep disturbances; and suicidal ideation increased before stay-at-home orders were imposed [33].

*Social provision:* Social support is conceptualized as “a process of perceiving the availability of different types of support through interpersonal relationships among social ties” [34]. Social provision, which comprises psychosocial factors, namely social relations and social support, is essential for the health and well-being of individuals [35]. Studies suggest that social support and social participation could have a positive impact on mental health and quality of life [34]. During the pandemic, people were advised to stay connected through social networking and virtual platforms [36]. They could still see facial expressions and hear voices, enjoying social interactions to alleviate depression and increase feelings of connectedness, while also observing social distancing [36].

*Family support:* Under the stay-at-home order, while observing social distancing, millions of people spent most of their time with friends and family because they were no longer spending time at work or attending school. These extraordinary conditions provide many people with opportunities to interact with their household members to fulfill their needs for closeness, belonging, and connection [37]. Parents, children, and household members could take advantage of extra time to engage in activities such as playing cards, board games, and watching movies together leading to deeper connections [36].

Given the discussion of the three concepts, we proposed a hypothesized structural equation (SEM) model of social connectedness, physical distancing and anxiety (SCPDA) to understand the association among the three concepts (Figure 1). The model consists of one latent variable and twelve observed variables. The dependent variable, days of physical distancing (X5), is influenced by socio-demographic variables, which include race, ethnicity, age, household size, number of people aged over 65 years, compliance with stay-at-home orders, employment status, and level of income (X6–X13, respectively). Social connectedness, a latent variable, is influenced by two measured variables, namely social provision (X1) and family support (X2). The social connected variable (X3) was influenced by the days of physical distancing (X5), whereas the two variables (X3 and X5) were moderated by the level of anxiety (X4). The following are hypothesized associations among the study variables.

**Hypothesis** **1.**
*Social provision (X1) is expected to be positively associated with social connectedness (X3).*


**Hypothesis** **2.**
*Family support (X2) is expected to be positively associated with social connectedness (X3).*


**Hypothesis** **3.**
*Social connectedness (X3) is expected to be positively associated with days of physical distancing (X5).*


**Hypothesis** **4.**
*Social connectedness (X3) is expected to be positively associated with level of anxiety (X4).*


**Hypothesis** **5.**
*Days of physical distancing (X5) is expected to be negatively associated with level of anxiety (X4).*


**Hypothesis** **6.**
*Race (non-White X6) is expected to be positively associated with days of physical distancing (X5).*


**Hypothesis** **7.**
*Ethnicity (Hispanic X7) is expected to be positively associated with days of physical distancing (X5).*


**Hypothesis** **8.**
*Age (X8) is expected to be positively associated with days of physical distancing (X5).*


**Hypothesis** **9.**
*Household size (number of people living at home X9) is expected to be positively associated with days of physical distancing (X5).*


**Hypothesis** **10.**
*Number of people with age 65 and older (X10) is expected to be positively associated with days of physical distancing (X5).*


**Hypothesis** **11.**
*Compliance to follow the stay-at-home order (X11) is expected to be positively associated with days of physical distancing (X5).*


**Hypothesis** **12.**
*Employment (status of currently working X12) is expected to be positively associated with days of physical distancing (X5).*


**Hypothesis** **13.**
*Level of income (X13) is expected to be positively associated with days of physical distancing (X5).*


## 2. Materials and Methods

This study followed the STROBE checklist for to report in order to improve the quality of reporting [38].

### 2.1. Sample

We conducted a nationwide online survey on the REDCap (Research Electronic Data Capture) platform, a secure web application for building and managing online surveys and databases between April and July 2020. During the 130 days period, there were 834 participants who were at least 18 years old. After excluding incomplete observations, this study included 494 samples with completed information on all measured variables. We did not compute the response rate because this study distributed the online survey links on different social media platforms and via email. As a result, we could not compute a traditional response rate. The resulting 494 participants are from 27 states across the nation (Figure 2B, Table A1 in Appendix C) with about 80% of them located in Texas. All the states where the participants were located had stay-at-home orders issued during the data collection period (Figure 2A,B, Figure A1 in Appendix A). During this period, the University of Texas Rio Grande Valley (UTRGV) published an invitation to participate in our survey via social media platforms. A large proportion of participants were chosen using a specific algorithm on social media that might have more frequently promoted social media posts to the people living in the State of Texas.

### 2.2. Measures

#### 2.2.1. Physical Distancing

Physical distancing was measured by the number of days that participants observed any type of stay-at-home orders and advisories during the study period. The total number of days was computed by subtraction between the beginning date and the end date showed by the participants. There were missing data on when the respondent began observing the stay-at-home order, and we treated these data by replacing the date of 1 March 2020) for two reasons. First, the participants set the earliest date as the beginning of stay-at-home was 20 February 2020. Second, people begun their staying at home even before the stay-at-home order was issued in all states because of the seriousness of the virus, which was emphasized in national and international news media [17]. The number of days were recoded into six groups: 1 (0–15 days), 2 (16–30 days), 3 (31–45 days), 4 (46–60 days), 5 (61–75 days) and 6 (76 days or more) (The survey instrument is shown in Appendix B).

#### 2.2.2. Social Connectedness

We measured social connectedness as a latent variable of the two related concepts, namely social provision and family support. The existing literature on social connectedness shows a limited consensus on concepts and methods to measure using evidence from ongoing debates [38,39]. Acknowledging the complexity of the concept, research suggests five dimensions that measure social connectedness [39]. They are closeness, identity and common bonds, valued relationships, involvement, and feeling cared for and accepted (giving the acronym CIVIC) [39]. In this study, social connectedness in the context of coping with the COVID-19 pandemic constitute two domains: social provision and family support. First, social provision was measured using the social provision scale (SPS), which comprises 10 items from five of the six dimensions of social support originally provided by Weiss [40]. The psychometric properties of the SPS10 items were discussed in previous studies [35,41,42]. The responses for each of the 10 items range from strongly agree (4), agree (3), disagree (2), to strongly disagree (1). The summation of the raw scores from the 10 items shows the degree of social provisions, where a high score indicates a high degree of social provision. The total scores are regrouped into mild (0–10), moderate (11–20), moderately high (21–30), and high (31–40) (Survey Instrument in Appendix B).

Second, we included a 12-item subscale: General Functioning (GF) of the McMaster Family Assessment Device (FAD). The validity of the subscale, as a single index that measures family function, was documented in previous studies [43,44,45]. The GF12 subscale comprises six items measuring the healthy aspects of family functioning and the other six capture unhealthy aspects. Each item is scored on a 4-point scale, from 1 being strongly disagree to 4 being strongly agree with healthy functioning, and the reverse order for items measuring unhealthy functioning. The total scores of the 12 items were categorized into 4 groups: 1 (0–10), 2 (1–20), 3 (21–30), and 4 (31 or more) (Survey Instrument in Appendix B).

#### 2.2.3. Anxiety

We used a Generalized Anxiety Disorder (GAD-7 scales), 7-item self-report measure to screen for the presence of generalized anxiety disorder. It was originally developed as a brief self-report scale to pre-screen probable cases of GAD [46], and its reliability and validity have been evaluated in many studies around the world. Of the 7 items, each has an answer ranging from 0 (not at all) to 3 (nearly every day), allowing for a maximum total score of 21, which could be interpreted as no/minimal anxiety (0–4), mild (5–9), moderate (10–14), or severe (15–21), with a cutoff point of 10 for GAD case (Survey Instrument in Appendix B).

We tested an inter-rater agreement for each of the three instruments that captured social provision, family support, and generalized anxiety disorder. Their agreement could be interpreted with a benchmark scale; <0.0000 for poor, 0.0000–0.2000 for slight, 0.2000–0.4000 for fair, 0.4000–0.6000 for moderate, 0.6000–0.8000 for substantial, and 0.8000–1.0000 for almost perfect agreement. The agreement coefficients are 0.7377 (substantial) for SPS-10 items, 0.3601 (moderate) for General Functioning (GF) of the McMaster Family Assessment Device (FAD), and 0.5779 (moderate) for GAD-7 scales.

#### 2.2.4. Socio-Demographic Variables

*Age* was measured with several years and divided into six group; 1 being (18–19 years old), 2 (20–29), 3 (30–39), 4 (40–49), 5 (50–59), and 6 (60 or older). *Race* was recoded as 0 for White and 1 for non-White (Black, Asian/Pacific Islander, American Indian/Aleut/Eskimo, and other). *Hispanic* was regrouped as 0 for non-Hispanic and 1 for Hispanic (Mexican, Mexican American, Chicano/a, Puerto Rican, Cuban, Another Hispanic, Latino/a, or Spanish origin). Family size or total number of people and the total number of people living with individuals aged over 65 years ranged from 1 to “6 or more” members. *Employment status* was recoded as 1 being “work” (Employed for wages and self-employed) and 0 being “no work” (Out of work for one year or more, Out of work for less than one year, Homemaker, Student, Retired, Unable to work, Other). *Income* was measured with 11 choices, 1 (USD 10,000 or less/less than USD 833 per month), 2 (USD 10,000-USD 19,999/USD 833-USD 1666 per month), 3 (USD 20,000-USD 29,999/ USD 1667-USD 2499 per month), 4 (USD 30,000-USD 39,999/USD 2500-USD 3332 per month), 5 (USD 40,000-USD 49,999 /USD 3333-USD 4166 per month), 6 (USD 50,000-USD 59,999/USD 4167-USD 4999 per month), 7 (USD 60,000-USD 69,999/USD 5000-USD 5833 per month), 8 (USD 70,000-USD 79,999/USD 5834-USD 6666 per month), 9 (USD 80,000-USD 89,999/USD 6667-USD 7500 per month), 10 (USD 90,000-USD 99,999/USD 7501-USD 8333 per month), and 11 (USD 100,000 and over /USD 8334 or over). *Compliance* to stay-at-home orders, shelter-in-place, or advisories was measured with a question of “I can safely stay in my home for at least 15 days” with a Likert Scale ranging from 1: Strongly Disagree to 4: Strongly Agree (Survey Instrument in Appendix B).

### 2.3. Structural Equation Modeling

To conduct a statistical examination of any associations between study variables, we proposed a hypothesized structural equation (SEM) model of social connectedness, physical distancing, and anxiety (SCPDA) (Figure 3). The model includes five endogenous variables: two observed variables (physical distancing and anxiety), two measurement variables (social provisions and family support), and one latent variable (social connectedness). It also comprises eight exogenous variables (observed): race, Hispanic, age, family size, individuals who are 65 years or older, compliance with stay-at-home orders and advisories, work status, and income. Our SCPDA model comprises both measurement and structural models; the first model includes social provisions, family support, and social connectedness. In our model, physical distancing (dependent variable), age, income and family size, which are measured in ordered categories of six, six, six, and eleven, respectively. We treated them as continuous variables in our model, and our decision is supported by studies demonstrating that such treatment did not have any impact on statistical analysis [47,48,49,50]. Given the discussion on our choice of treating the study variables, we selected the maximum likelihood (ML) method to run our model.

## 3. Results

### 3.1. Descriptive Statistics

Results of the descriptive statistics were presented in Appendix C, Table A2. The total respondents (494) comprise 87% White; 85% Hispanic; 80% female; 46% aged between 20 and 39; 37% with 2 children; 15% living with one elderly person aged older than 65; 82% with high family support; 78% with very high social provision; 48% with mild anxiety; 63% in strong compliance with stay-at-home orders; 76% working; and 38% with an income of USD 100,000 or higher.

### 3.2. SEM Results

To evaluate our hypothesized associations among the study variables (Table A5, Appendix C), we first conducted the structural equation analysis of the SCPDA model using the ML estimation method. The overall goodness-of-fit results (χ^2^ (26) = 49.612, CFI = 0.823, TLI = 0.731, RMSEA = 0.045) show that the model is a relatively good fit (Table A4 in Appendix C). We also checked both the mean and covariance residuals of the observed variables, which had either small values or were zero, also showing a relatively good fit (Table A3 in Appendix C).

Firstly, the results show that social provisions (*b* = −0.256, *p* < 0.000) are negatively associated with social connectedness (Table 1, Figure 3). In other words, a decrease in social provision is associated with an increase in social connectedness. Secondly, as expected, family support (*b* = 0.149, *p* < 0.004) is positively associated with social connectedness (Table 1, Figure 3). It could be interpreted that an increase in family support is positively associated with an increase in social connectedness. Observing stay-at-home recommendations requires all family members in a household to stay together. The more days the respondents spend together in their home, the more they understand and support one another, creating more social connectedness among the members. Thirdly, the results show that anxiety (*b* = −1.781, *p* < 0.001) was negatively associated with physical distancing (Table 1, Figure 3). The higher the number of days respondents spent observing stay-at-home orders, the lower level of anxiety they experienced. In other words, at the beginning of their time spent at home, participants were likely to experience some anxiety. However, the longer they stayed at home, the lower their anxiety became. Fourthly, an increase in anxiety (*b* = 1.054, *p* < 0.000) was observed with an increase in social connectedness (Table 1, Figure 3). Fifthly, social connectedness (b = −1.398, *p* < 0.000) showed a positive association with the physical distancing. In other words, participants who experienced a higher level of social connectedness are likely to observe physical distancing for longer (Table 1, Figure 3). Sixthly, a decrease in income (*b* = −0.309, *p* < 0.011) showed an increase in physical distancing. Notably, respondents whose income is lower did observe stay-at-home orders for longer (Table 1, Figure 3). Seventhly, the age of respondents (b = −0.414, *p* < 0.003) showed a negative association with the physical distancing. In other words, respondents who are younger in age are likely to experience observe physical distancing for a shorter time, while their older counterparts are likely to spend longer observing physical distancing (Table 1, Figure 3). Eighthly, respondents whose families have many members (b = 0.207, *p* < 0.034) showed a positive association with physical distancing (Table 1, Figure 3).

## 4. Discussion

Our SCPDA model comprises three important concepts, namely, social connectedness, anxiety, and social distancing.

First, social connectedness shows a positive association with family support and a negative association with social provision. This could imply that social interactions among parents, children, and household members could lead them to deeper connections. Secondly, anxiety shows a negative association with physical distancing, whereas it has a positive association with social connectedness. These findings suggest that the longer respondents observed physical distancing, the more anxiety they felt. In addition, regarding social connectedness, the respondents still felt some anxiety given that they had to physically distance from others. It could be implied that this anxiety could not be eliminated by social connectedness, which could be provided by family members. Third, physical distancing shows a positive association with social connectedness and a negative association with anxiety. This suggests that social connectedness could lead to an ability to observe physical distancing for longer. However, physical distancing could influence the feelings of anxiety among the respondents. In addition, the three main drivers that strongly influence physical distancing are income, family size, and age. Most respondents were younger, had high incomes, and were from small families.

### Caveats and Limitations

There are three limitations to this study. Firstly, despite the nationwide invitation for participants, about 80% of the respondents lived in Texas. When interpreting the findings and making generalizations, this sample distribution should be cautiously taken into account. Secondly, this survey was conducted online, which might unintentionally exclude those who do not have access to an internet connection. Thirdly, the investigation period of this study was at the beginning of the COVID-19 pandemic when vaccinations were not yet available. Possible changes after the sampling period were not taken into account in this study.

## 5. Conclusions

In this study, we proposed our SCPDA model, which conceptualizes the relationships between three main domains: social connectedness, anxiety, and physical distancing. A national survey with 494 respondents was conducted online; the majority of them are from Texas. Social connectedness was positively influenced by family support, whereas a negative association was found with the social provision. This finding suggests that, while observing the stay-at-home order, family support plays an important role in feeling socially connected during the pandemic [51,52,53]. It is possible that many of the respondents might spend much time with their family members and engage in activities that help to strengthen social connections during the pandemic, while they might have less time to interact with their friends in public spaces. Secondly, social connectedness has a positive association with anxiety, which is contrary to findings in some studies that show a negative association [51,54]. It is possible that the anxiety stemmed from both a lack of social connectedness and the fear of potential infection during the pandemic. Although, social connectedness could help reduce some anxiety, the anxiety and fear of becoming infected with coronavirus due to the unavailability of pharmaceutical interventions, such as COVID-19 vaccines, prevailed during the study period [55,56]. The model shows a positive association between social connectedness and physical distancing. The association suggests two implications: During the pandemic, promoting social connectedness could result in the desired outcome of physical distancing, and the use of term social distancing is not appropriate [57]. Among the tested socio-economic characteristics, the factor of income influenced physical distancing. This finding is in line with the results of another study in the United States showing that lower-income neighborhoods could not observe physical distancing because of the necessity to work outside [58].

## Figures and Tables

**Figure 1 ijerph-19-15282-f001:**
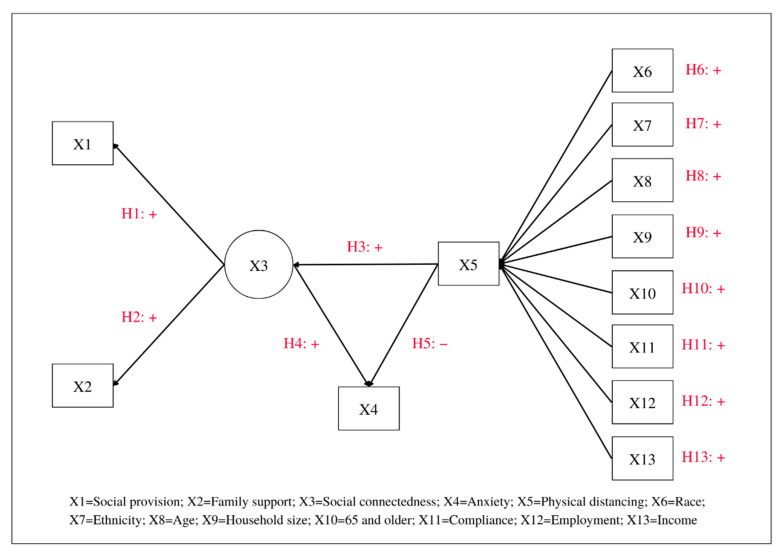
A Hypothesized Structural Equation (SEM) Model of Social Connectedness, Physical Distancing, and Anxiety (SCPDA).

**Figure 2 ijerph-19-15282-f002:**
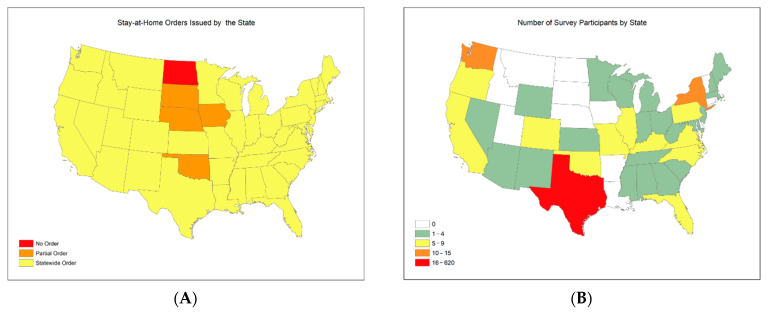
Stay-at-home orders issued state and geographical distributions of survey participants. (**A**). Stay-at-home orders issued by states between March and May 2020. (**B**). Geographical distribution of survey participants across US.

**Figure 3 ijerph-19-15282-f003:**
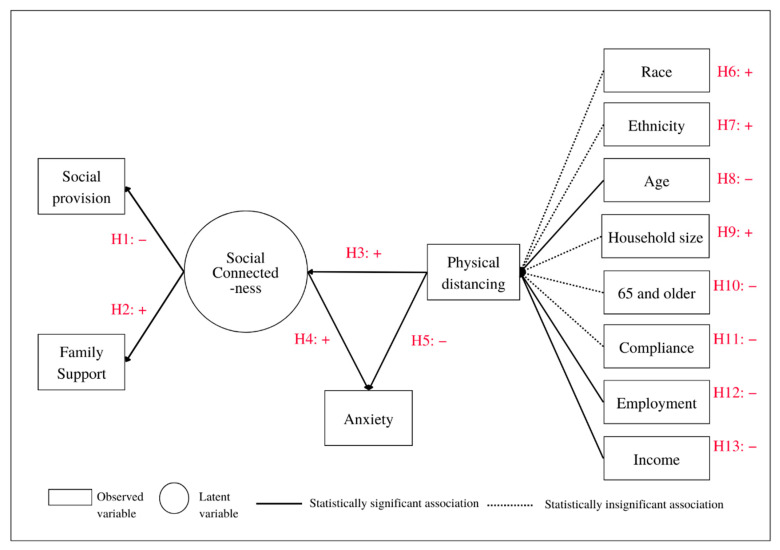
Validation of associations among study variables in the hypothesized structural equation model (SEM) of social connectedness, physical distancing, and anxiety (SCPDA).

**Table 1 ijerph-19-15282-t001:** Results of multivariate regression analysis using structural equation (SEM) model.

	Standardized		Standard		
	Coefficient		Error	z	*p* > z
**Structural**					
Physical distancing					
Anxiety	−1.640	*	0.477	−3.440	0.001
Race	0.156		0.085	1.850	0.065
Hispanic	0.079		0.091	0.870	0.385
Age	−0.414	*	0.141	−2.950	0.003
Family size	0.207	*	0.098	2.120	0.034
65 years and older	−0.072		0.095	−0.760	0.448
Compliance	−0.195		0.100	−1.950	0.052
Employment status	−0.111		0.093	−1.200	0.231
Income	−0.309	*	0.121	−2.540	0.011
Constant	7.824	***	1.631	4.800	0.000
Anxiety					
Social connectedness	1.054	***	0.019	55.590	0.000
Constant	−1.670		0.848	−1.970	0.049
Social connectedness					
Physical distancing	1.398	***	0.318	4.400	0.000
**Measurement**					
Social provisions					
Social connectedness	−0.255	***	0.053	−4.760	0.000
Constant	7.585	***	0.320	23.720	0.000
Family support					
Social connectedness	0.148	***	0.052	2.830	0.005
Constant	7.707	***	0.348	22.180	0.000
var(e.Physical distancing)	3.418		1.428	1.507	7.752
var(e.Anxiety)	0.266		0.279	0.034	2.084
var(e.Social provisions)	0.935		0.029	0.880	0.994
var(e.Family support)	0.978		0.016	0.947	1.010
var(e.Social connectedness)	2.303		0.820	1.146	4.627

N = 494, LR test of model vs. saturated: chi2(25) = 49.61, Prob > chi2 = 0.0024, * *p* < 0.05, *** *p* < 0.001.

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
