# Peer review of "Social Connectedness, Physical Distancing, and Anxiety in Complying with Shelter-In-Place Orders and Advisories during the Once-In-A-Century COVID-19 Pandemic in the US: A Study of Social Media and Internet Users"

_ijerph, 2022, doi:10.3390/ijerph192215282_

Round 1
Reviewer 1 Report
The paper presents a hypothesized structural model of social connectedness, physical distancing and anxiety (SCPDA) to investigate the association among the three concepts. Data collection was conducted through online questionnaires, resulting in 494 respondents, mainly from Texas. The introduction provides sufficient details to describe to scientific question, and also the methods section is clear for the reader. Results, with declared limitations about the coverage of data that limits the generalization of the findings, are presented and discussed exploiting also Appendixes. References are adequate.
The starting Hypothesis provided in the Introduction should be described in more detail, trying to explain what they are based on. Moreover, Hyp. 2, 3 and 4 are identical.
Overall, references to figures and tables need to be revised.
Finally, it is not clear for the reviewer why this work prepared in October 2020 is proposed for publication only after 2 years.
Author Response
Good morning,
First of all, we would like to express appreciation for reviewing our manuscript.
Our response to your comments will appear under each of your paragraphs.
The paper presents a hypothesized structural model of social connectedness, physical distancing and anxiety (SCPDA) to investigate the association among the three concepts. Data collection was conducted through online questionnaires, resulting in 494 respondents, mainly from Texas. The introduction provides sufficient details to describe to scientific question, and also the methods section is clear for the reader. Results, with declared limitations about the coverage of data that limits the generalization of the findings, are presented and discussed exploiting also Appendixes. References are adequate.
The starting Hypothesis provided in the Introduction should be described in more detail, trying to explain what they are based on. Moreover, Hyp. 2, 3 and 4 are identical.
Overall, references to figures and tables need to be revised.
Finally, it is not clear for the reviewer why this work prepared in October 2020 is proposed for publication only after 2 years.
“The paper presents a hypothesized structural model of social connectedness, physical distancing and anxiety (SCPDA) to investigate the association among the three concepts. Data collection was conducted through online questionnaires, resulting in 494 respondents, mainly from Texas. The introduction provides sufficient details to describe to scientific question, and also the methods section is clear for the reader. Results, with declared limitations about the coverage of data that limits the generalization of the findings, are presented and discussed exploiting also Appendixes. References are adequate.”
Thank you for your constructive remarks.
“The starting Hypothesis provided in the Introduction should be described in more detail, trying to explain what they are based on. Moreover, Hyp. 2, 3 and 4 are identical.”
Yes. They have been revised.
“Overall, references to figures and tables need to be revised.”
Thank you. They have been revised.
“Finally, it is not clear for the reviewer why this work prepared in October 2020 is proposed for publication only after 2 years.”
Thank you for asking this question. At the outset of our collaborative research, we planned to submit this manuscript to this prestigious journal, International Journal of Environmental Research and Public Health. However, we could not secure financial resources when our paper was completed. We tried many ways to secure the fund, and finally we received from one of our affiliated departments. As soon as the funds became available, we submitted this manuscript for review process. We hope that our revision goes well and will receive an opportunity to publish our paper in this journal.
Thank you so much. Please do accept our hearty appreciation.
Kind regards,
Authors
Reviewer 2 Report
The authors aim to examine the relationship between social connectedness and anxiety, among individuals living under the COVID-19 related social distancing conditions, shelter-in-place compliance and advisories during the pandemic. This is very interesting and very much needed study.
The theoretical overview of the physical distancing literature is well presented. On the other hand, social connectedness literature section lacks an acknowledgment of the conceptual and methodological debate in the literature on social connectedness. There is very little agreement on what social connectedness is and how it can be measured. Some of the examples of the literature to review and consider are:
Hare-Duke, L., Dening, T., de Oliveira, D., Milner, K. and Slade, M., 2019. Conceptual framework for social connectedness in mental disorders: Systematic review and narrative synthesis. Journal of affective disorders, 245, pp.188-199.
Valtorta, N.K., Kanaan, M., Gilbody, S. and Hanratty, B., 2016. Loneliness, social isolation and social relationships: what are we measuring? A novel framework for classifying and comparing tools. BMJ open, 6(4), p.e010799.
Most importantly, the authors do not elaborate on why they have chosen social provision and family support as measures of social connectedness.
Also, there seems to be some grammatical errors in the social connectedness section “The association between social disconnectedness and physical and 193 mental health outcome is highly coupled”
Similarly, in the conclusion section, some of the sections are not clear “These findings suggest that during the pandemic, social distancing is not encouraged, but more social connectedness while observing physical distancing”
In regards to methods, I am not too familiar with the SEM. 14 hypotheses, as listed in the paper, are difficult to follow. I suggest that the authors present the hypotheses in a more visually useful way, similar to what they have in Figure 3. Ex Age, Household Size are indicators of X; employment, family size are indicators of ….; The effect of family support on X is moderated by ….; etc.
Most importantly, it is not clear how Hypothesis 2, 3 and 4 differ. They seem exactly the same.
These issues should addressed before a more detailed commentary on the paper results could be provided.
Author Response
Good morning,
First of all, we would like to express appreciation for reviewing our manuscript.
Our response to your comments will appear under each of your paragraphs.
The authors aim to examine the relationship between social connectedness and anxiety, among individuals living under the COVID-19 related social distancing conditions, shelter-in-place compliance and advisories during the pandemic. This is very interesting and very much needed study.
Thank you so much for your kind remarks.
The theoretical overview of the physical distancing literature is well presented. On the other hand, social connectedness literature section lacks an acknowledgment of the conceptual and methodological debate in the literature on social connectedness. There is very little agreement on what social connectedness is and how it can be measured. Some of the examples of the literature to review and consider are:
Hare-Duke, L., Dening, T., de Oliveira, D., Milner, K. and Slade, M., 2019. Conceptual framework for social connectedness in mental disorders: Systematic review and narrative synthesis. Journal of affective disorders, 245, pp.188-199.
Valtorta, N.K., Kanaan, M., Gilbody, S. and Hanratty, B., 2016. Loneliness, social isolation and social relationships: what are we measuring? A novel framework for classifying and comparing tools. BMJ open, 6(4), p.e010799.
Again, thank you for your suggestion and comment on this particular area of social connectedness. We included additional literature, based on your suggested two articles, which we found relevant and very interesting.
Most importantly, the authors do not elaborate on why they have chosen social provision and family support as measures of social connectedness.
We included an additional description which provided reasons for selecting social provision and family support as measures of social connectedness in this specific context of coping with the pandemic.
Also, there seems to be some grammatical errors in the social connectedness section “The association between social disconnectedness and physical and 193 mental health outcome is highly coupled”
We fixed it by rewriting the sentence. “A strong association between social disconnectedness and physical and mental health outcome was observed.” Thank you.
Similarly, in the conclusion section, some of the sections are not clear “These findings suggest that during the pandemic, social distancing is not encouraged, but more social connectedness while observing physical distancing”
We rewrote the sentence. “These findings suggest two things; during the pandemic, promoting social connectedness could result in a desired outcome of physical distancing; and the use of term social distancing is not appropriate.”
In regards to methods, I am not too familiar with the SEM. 14 hypotheses, as listed in the paper, are difficult to follow. I suggest that the authors present the hypotheses in a more visually useful way, similar to what they have in Figure 3. Ex Age, Household Size are indicators of X; employment, family size are indicators of ….; The effect of family support on X is moderated by ….; etc.
We provided new revised figures for 1 and 3.
Most importantly, it is not clear how Hypothesis 2, 3 and 4 differ. They seem exactly the same.
We have corrected the hypothesis 2, 3, and 4.
These issues should addressed before a more detailed commentary on the paper results could be provided.
Thank you so much.
Thank you so much. Please do accept our hearty appreciation.
Kind regards,
Authors
Reviewer 3 Report
suggest the Editor to accept the manuscript after some changes.
Comments to Author(s):
Thanks for this interesting study. I have comments for the Authors that I hope they consider as opportunity to improve their research.
The investigation period in at the beginning of the Covid-19 Pandemic, and maybe some aspects could be changed in three years, or not?
The Authors explain well the choice of the main topics investigated: the social connectedness, physical distancing and anxiety aspects are considered as key points for the good compliance of mandatory lockdown advisories ì (stay at home).
I suggest to explain better the strategy used to disseminate / divulgate the on line survey to the participants (which media platform, channels, main list, etc.).
The sample size is really low for generalizability of the results for the US states and it is correctly underlined from the Authors. The research shown a majority of the participants from Texas. Is it possible to study examine only Texas participants? It is possible to explain why there is so high different percentage of Texan participants to the survey? The dissemination of the survey in Texas has presented different strategy?
Other aspect that I would underlines is the followed. The responders are who have familiarity and possibility to use internet and it is could be introduce a selection bias. This aspect is declared from the Authors, too. But, I think that it is not a limits; it is a peculiarity of the study. If the data collection was defined in the study protocol through the use of an on-line survey, the Authors have decided that their targets population will be who can fill an on-line questionnaire.
So it is important to consider in the methods, that the target population is who have opportunity to fill-in an on-line questionnaire and have the ability to access to the social media platforms, to use social networks, to consult e-mails, etc. And with this in mind, maybe you can reconsider the title of the research.
I recommend to the Authors to follow the STROBE STATEMENT and complete their check when it is possible.
Author Response
Good morning,
First of all, we would like to express appreciation for reviewing our manuscript.
Our response to your comments will appear under each of your paragraphs.
Thanks for this interesting study. I have comments for the Authors that I hope they consider as opportunity to improve their research.
Thank you for your kind remarks. We appreciated much.
The investigation period in at the beginning of the Covid-19 Pandemic, and maybe some aspects could be changed in three years, or not?
It is a good point. We have incorporated this limitation under the Caveats and Limitation of the study.
The Authors explain well the choice of the main topics investigated: the social connectedness, physical distancing and anxiety aspects are considered as key points for the good compliance of mandatory lockdown advisories ì (stay at home).
Thank you for your kind remarks.
I suggest to explain better the strategy used to disseminate / divulgate the on line survey to the participants (which media platform, channels, main list, etc.).
We included an additional explanation on survey distribution.
The sample size is really low for generalizability of the results for the US states and it is correctly underlined from the Authors. The research shown a majority of the participants from Texas. Is it possible to study examine only Texas participants? It is possible to explain why there is so high different percentage of Texan participants to the survey? The dissemination of the survey in Texas has presented different strategy?
We estimated the sample size for the population of US who are 18 years or above (209,128,094) with margin of error 5%, and confidence level 95 using a sample size calculator, which provided as 385. Our sample size (494) exceeded the estimated sample size. We reported our sample which comprised 80% respondents from the State of Texas. In this revision, we also added explanation on the presence of larger portion of the participants from Texas. “During the data collection period, the University of Texas Rio Grande Valley (UTRGV) published invitation to participate in our survey via social media platforms. However, our speculation was that the algorithm in the social media might have reached out more frequently to the people in Texas than they did with other states.”
Other aspect that I would underlines is the followed. The responders are who have familiarity and possibility to use internet and it is could be introduce a selection bias. This aspect is declared from the Authors, too. But, I think that it is not a limits; it is a peculiarity of the study. If the data collection was defined in the study protocol through the use of an on-line survey, the Authors have decided that their targets population will be who can fill an on-line questionnaire. So it is important to consider in the methods, that the target population is who have opportunity to fill-in an on-line questionnaire and have the ability to access to the social media platforms, to use social networks, to consult e-mails, etc. And with this in mind, maybe you can reconsider the title of the research. I recommend to the Authors to follow the STROBE STATEMENT and complete their check when it is possible.
This is an excellent point. We appreciated your thoughtfulness on this study population and suggesting the checklist of STROBE STATEMENT. After following your advice and checking the checklist of STROBE STATEMENT, an additional phrase was added to the topic to reflect predetermined population who has an internet access to use emails and social medias. Our title has 29 words. We could not find any limitation on the title length on the journal’s page.
“Social Connectedness, Physical Distancing and Anxiety in Complying with Shelter-in-Place Orders and Advisories during the Once-in-a-Century COVID-19 Pandemic in the US: A Study of Social Media and Internet Users.”
Thank you so much. Please do accept our hearty appreciation.
Kind regards,
Authors
Round 2
Reviewer 2 Report
The authors have put efforts into addressing comments. However, there are still a number of sentences in the paper that seem not to be grammatically correct. Ex "The existing literature in social connectedness shows limited agreement on conceptual and methods to measure with evident of 318 ongoing debate"
The authors are advised to identify and address grammatical issues in the paper.
Author Response
Reviewer 2
Open Review
(x) I would not like to sign my review report
( ) I would like to sign my review report
English language and style
( ) English very difficult to understand/incomprehensible
( ) Extensive editing of English language and style required
(x) Moderate English changes required
( ) English language and style are fine/minor spell check required
( ) I don't feel qualified to judge about the English language and style
|
Yes |
Can be improved |
Must be improved |
Not applicable |
|
|
Does the introduction provide sufficient background and include all relevant references? |
(x) |
( ) |
( ) |
( ) |
|
Are all the cited references relevant to the research? |
( ) |
(x) |
( ) |
( ) |
|
Is the research design appropriate? |
(x) |
( ) |
( ) |
( ) |
|
Are the methods adequately described? |
( ) |
(x) |
( ) |
( ) |
|
Are the results clearly presented? |
(x) |
( ) |
( ) |
( ) |
|
Are the conclusions supported by the results? |
( ) |
(x) |
( ) |
( ) |
Comments and Suggestions for Authors
Good morning,
First of all, we would like to express appreciation for reviewing our revised version.
Our response to your comments will appear under each of your paragraphs.
The authors have put efforts into addressing comments. However, there are still a number of sentences in the paper that seem not to be grammatically correct. Ex "The existing literature in social connectedness shows limited agreement on conceptual and methods to measure with evident of 318 ongoing debate"
The authors are advised to identify and address grammatical issues in the paper.
As suggested, we have undergone an extensive English revision with the service provided by the English Language Editing by the IJERPH. I hope that our edited version has improved and meets your expectations. We truly appreciated all the detailed comments and suggestions you have made to improve our manuscripts. We also reviewed references and removed and replaced with the references that are relevant to the context in the US.
Thank you so much. Please do accept our hearty appreciation.
Kind regards,
Authors
Submission Date
03 October 2022
Date of this review
09 Nov 2022 11:39:20

Reviewer 3 Report
The Authors have declared to apply he STROBE statement, so I recommend to the Authors to cite the STROBE statement at the beginning of the "Methods" . For example see doi: 10.1016/j.jclinepi.2007.11.008.
Author Response
Reviewer 3
Open Review
( ) I would not like to sign my review report
(x) I would like to sign my review report
English language and style
( ) English very difficult to understand/incomprehensible
( ) Extensive editing of English language and style required
( ) Moderate English changes required
( ) English language and style are fine/minor spell check required
(x) I don't feel qualified to judge about the English language and style
|
Yes |
Can be improved |
Must be improved |
Not applicable |
|
|
Does the introduction provide sufficient background and include all relevant references? |
(x) |
( ) |
( ) |
( ) |
|
Are all the cited references relevant to the research? |
(x) |
( ) |
( ) |
( ) |
|
Is the research design appropriate? |
(x) |
( ) |
( ) |
( ) |
|
Are the methods adequately described? |
( ) |
(x) |
( ) |
( ) |
|
Are the results clearly presented? |
(x) |
( ) |
( ) |
( ) |
|
Are the conclusions supported by the results? |
(x) |
( ) |
( ) |
( ) |
Comments and Suggestions for Authors
Good morning,
First of all, we would like to express appreciation for reviewing our manuscript.
Our response to your comments will appear under each of your paragraphs.
The Authors have declared to apply he STROBE statement, so I recommend to the Authors to cite the STROBE statement at the beginning of the "Methods" . For example see doi: 10.1016/j.jclinepi.2007.11.008.
Please accept my apologies for oversight. We have added this statement at the beginning of the “Methods.”
Thank you so much. Please do accept our hearty appreciation.
Kind regards,
Authors
Submission Date
03 October 2022
Date of this review
07 Nov 2022 13:47:00
